# Projections of the Net Primary Production of Terrestrial Ecosystem and Spatiotemporal Responses to Climate Change in the Yangtze River Economic Belt

Li Yu [1], Botao Zhou [2], Yuqing Xu [1], Yongxiang Zhang [1] and Fengxue Gu [3],*

1 National Climate Center, China Meteorological Administration, Beijing 100081, China; yuli@cma.cn (L.Y.); xuyq@cma.cn (Y.X.); zhangyx@cma.cn (Y.Z.)
2 Collaborative Innovation Center on Forecast and Evaluation of Meteorological Disasters, Key Laboratory of Meteorological Disaster, Ministry of Education, Joint International Research of Climate and Environment Change, Nanjing University of Information Science and Technology, Nanjing 210044, China; zhoubt@nuist.edu.cn
3 Key Laboratory of Dryland Agriculture, Institute of Environment and Sustainable Development in Agriculture, Chinese Academy of Agriculture Sciences, Beijing 100081, China
* Correspondence: gufengxue@caas.cn

**Abstract:** Evaluating the responses of net primary productivity (NPP) to climate change is essential for regional ecosystem management and adaptations to climate change. The Yangtze River Economic Belt (YREB) is a key ecological functional area and hotspot of carbon sequestration in China due to the high degree of forest coverage. We used a process-based ecosystem model to project terrestrial NPP and analyzed the response to climate change over the 21st century in the YREB under two representative concentration pathway (RCP) scenarios using the regional climate model. The results show that the projection of NPP generally increased by 13.5% under RCP4.5 and 16.4% under RCP8.5 in the middle of the century, by 23% under RCP4.5, and by 35% under RCP8.5 in the late term of the century compared with that from a reference period of current climate conditions (1985–2006). The rate of NPP change under the RCP8.5 scenario is higher than that under the RCP4.5 scenario. Similarly, the NPP is also projected to increase both with 1.5 and 2 °C global warming targets in the YREB. The magnitudes of NPP increment are approximately 14.7% with 1.5 °C and 21% with 2 °C warming targets compared with the current climate, which are higher than the average increments of China. Although NPP is projected to increase under the two scenarios, the tendency of NPP increasingly exhibits a slowdown after the 2060 s under the RCP4.5 scenario, and the growth rate of NPP is projected to drop in more than 31% of regional areas with the additional 0.5 °C warming. In contrast, under the RCP8.5 scenario, the trend in NPP keeps rising substantially, even above 2 °C global warming. However, the NPP in some provinces, including Jiangxi and Hunan, is projected to reduce at the end of the 21st century, probably because of temperature rises, precipitation decreases, and water demand increases. Generally, the NPP is projected to increase due to climate change, particularly temperature increase. However, temperature rising does not always show a positive effect on NPP increasing; the growth rate of NPP will slow down under the RCP4.5 scenario in the mid-late 21st century, and NPP will also reduce by the end of this century under the RCP8.5 scenario in some places, probably presenting some risks to terrestrial ecosystems in these areas, in terms of reduced functions and service decline, a weakened capacity of carbon sequestration, and reduced agricultural production.

**Keywords:** vegetation NPP; projection; process-based ecosystem model; climate change; global warming target; YREB

## 1. Introduction

Terrestrial ecosystems play a key role in the global carbon cycle, which absorbs approximately 20% and 30% of anthropogenic $CO_2$ emissions worldwide [1]. The net primary

productivity (NPP) of the terrestrial ecosystems is a critical indicator of the terrestrial carbon cycle, as well as being an essential component for the survival of ecosystems that contribute to ensuring the welfare of human beings [2,3]. Exploring NPP responses to climate change and its feedback is crucial for understanding the terrestrial ecosystem dynamics and sustainable development. Terrestrial ecosystem models represent a common framework and a useful method for the ecological research of climate processes, which provide simulations not just for determining the outcomes of terrestrial ecosystems, but also assessing the responses to future stresses [4]. Previous studies have confirmed that the terrestrial NPP is highly sensitive to climate change and other environmental factors [5,6]. Continuous warming has profound effects on the NPP of terrestrial ecosystems, as shown by observational evidence and modeling findings, demonstrating extensive influences on ecosystem functions and services [1,7]. On the global scale, the gross primary productivity (GPP) and NPP have provided evidence of changes, and are projected to increase or remain unchanged, especially in mid-to-high-latitude areas [8,9]. However, on regional scales, the estimations of NPP have varied widely among different studies, and the trends in NPP have exhibited more divergence in different climatic zones [10,11]. Some studies based on ecosystem models and remote sensing data found that the NPP increased on a national scale or within certain typical ecosystem types in recent decades [12,13]; other studies found that the trends of NPP deceased over time in some regions [14]. However, estimations of average NPP ranged considerably on regional scales [15–17]. Furthermore, the projections of NPP showed more inconsistencies in terms of magnitude, trends, and spatial distribution, and were sometimes even controversial when projecting the responses of NPP to future climate change [14,18]. Exploring the spatiotemporal patterns of NPP and dominant factors concretely can develop an understanding about terrestrial carbon sequestration and ecological risks to climate change on different scales; this is also required for facilitating climate change adaptation and ecosystem management.

The Yangtze River is the third largest river in the world, and its development is central to one of three ongoing national strategies of the One Belt and One Road initiative in China, which a project of high international significance. However, it is also one of the regions most sensitive to climate change, being deeply affected by the East Asian Monsoon [19]. The annual average temperature of the Yangtze River Basin has increased more than the global average. Furthermore, the average regional temperature is projected to rise remarkably in the future according to the projections of climate models, as will the frequency of the extreme climatic events occurrence [20,21]. On the other hand, this region has the largest subtropical forest in the world, and the carbon sequestration of artificial forests is considered to represent huge potential for achieving China's carbon neutral target, due to the sufficient water and thermal resources [22,23]. However, a range of ecological and environmental problems, such as the shrinking area of natural forests, ecosystem degradation, soil erosion, reductions in biodiversity, and rapid urbanization, have emerged in the Yangtze River Basin since the 1950s [24,25]. To protect the fundamental ecological barriers and a major grain-producing region of China, a series of guidelines have been issued successively, e.g., "Grain for Green" (since 1990s), the "Outline for the Development Plan of the Yangtze River Economic Belt (YREB)" (2016), and the "Environmental Protection Plan for the YREB" (2017). The future of the YREB is proposed to orient towards ecological priority and green development [26].

With the green development of the YREB clearly confirmed, more and more studies have focused on changes in the ecosystems and its responses to environmental factors. Ke et al. used the Carnegie–Ames–Stanford Approach (CASA) model to simulate the spatial and temporal patterns of vegetation NPP in the Yangtze River Basin from 1982 to 1999, and found that it exhibited a clear increasing tendency of NPP [27]. Zhang et al. estimated that the forest NPP decreased from southeast to northwest by the Lund–Potsdam–Jena (LPJ) model in 1981–2000 in the Yangtze River Basin [28]. However, the estimates of NPP varied in magnitude and pattern, ranging from 262 to 687 g C/m$^2$ a$^1$, because of differences in approaches, data, time frames, etc. [29,30]. As for the NPP changes in the future, research by

Miao et al. suggested that terrestrial NPP will decrease in most of the Yangtze River Basin under SRES B2 scenarios using the Atmosphere–Vegetation Interaction Model (AVIM) [30]. Other studies in China also suggested that the Yangtze River Basin is at high ecological risk due to the decrease in ecosystem productivity caused by climate change [31]. However, some studies on the region had projected an increasing tendency of terrestrial NPP [11,18]. In summary, the results of existing studies indicated that there are considerable variations and high uncertainties, whether estimating NPP in recent decades or in projections of NPP in the future in the YREB. According to the implementation of a new regional development philosophy that began in the 1980s, it is also necessary to track the effects of ecosystem programs and understand the dynamics of carbon sequestration in this region, exploring the ecosystem responses and risks associated with climate change in the future.

In this study, terrestrial ecosystem NPP was estimated and projected by a process-based ecosystem model with high-resolution data of vegetation types and Regional Climate Model (RCM) results in 1971–2099, to explore the responses of the ecosystem to climate change in the YREB, which is a significant river basin that covers diverse terrestrial ecosystems in a typical subtropical monsoon region. The major objectives of this study are: (1) to project the spatiotemporal patterns of terrestrial ecosystem NPP in the 21st century, (2) to investigate the changing trends in NPP over different periods and with different global warming targets; and (3) to explore the responses of NPP and potential risks to climate change in the YREB.

## 2. Materials and Methods

### 2.1. Study Area

The YRBE is located in the middle of China, and covers nine provinces and two municipalities, over an area of 2.05 billion square kilometers, accounting for 21% of China's landmass (Figure 1). The YREB has a very important ecological role in China, as well as for the rest of East Asia. Approximately 40% of China's available fresh-water resources and more than one-fifth of the total wetland area are here [32]. Grain production, water conservation, and carbon sequestration in the YREB account for 40.2%, 39.2%, and 37.7% of China's total proportions for terrestrial ecosystem services, respectively [33]. Subtropical forests constitute approximately 45% of this region, which is considered an important carbon sink in the Northern Hemisphere [23].

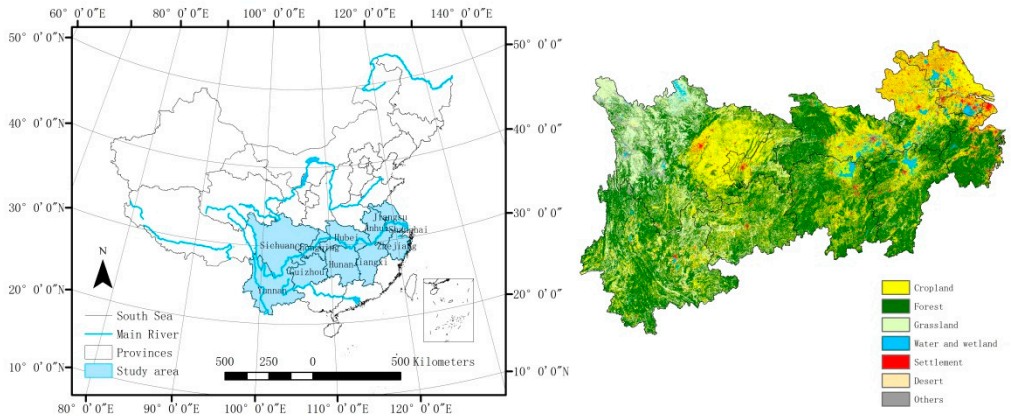

**Figure 1.** The geographical location and ecological pattern of the YREB of China.

### 2.2. Model Description and Validation

The Carbon Exchange between Vegetation, Soil, and Atmosphere (CEVSA) model is a process-based terrestrial ecosystem mechanical model that simulates and quantifies energy and the cycles of carbon, nitrogen and water [34]. There are three key modules in the CEVSA model: the biophysical module, which calculates the canopy conductance of plant, soil water supply, and evapotranspiration; the plant growth module, including photosynthesis, autotrophic respiration, carbon allocation in organs, etc., which are key

processes of plant growth; and the biochemical module, which simulates how carbon and nitrogen transform and decompose between vegetation and soil. The major parameters and processes of the CEVSA model have been validated and calibrated in different ecosystem types. Gross primary productivity (GPP), NPP, ecosystem heterotrophic respiration (HR), leaf area index (LAI), and net ecosystem exchange outputs by the CEVSA model have exhibited high correlations with the observations by eddy flux tower data and plot-sampling observation datasets of the China Ecosystem Research Network, and the value of remote sensing estimated models [35,36]. The CEVSA model has been widely used to simulate the responses of ecosystems to environmental factors on different scales [18,37]. In this study, we used the latest vision of the CEVSA model—CEVSA2, which incorporates a nitrogen influence module and updated data on nitrogen depositions. The details of the CEVSA2 model are described in Gu et al. (2017) [36]. Comparisons between the observed data of vegetation carbon storage and soil carbon storage showed that the simulation results of the improved CEVSA2 model have stronger agreement with the observed data [37].

*2.3. Input Datasets*

The gridded datasets included vegetation types, soil parameters, and climate processes, and were inputted to the CEVSA2 model. All input datasets were processed to 10-day averages and interpolated to 0.1 latitude × 0.1 longitude resolutions. Climate data were interpolated with the thin disk spline algorithm in ANUSPLIN software [38]. Vegetation types derived from Global Land Cover (GLC) data were downloaded from the website of the European Commission and resampled to 16 vegetation function types for the CEVSA model. The soil data and parameters were derived from a digitalized soil texture map of China and a soil classifications map, provided by the Institute of Soil Science, Chinese Academy of Science. Climate input data were used for the period of 1961–2099, which constituted observational data for the simulation of ecosystem responses under current climate condition, and climate change scenario data were used to project further ecosystem responses. The observational data were based on daily dataset in situ measurements from about 2400 meteorological stations from 1961 to 2005, which were provided by National Meteorological Information Center. The climate change scenario data were obtained from the BCC_CSM1.1 global climate model, driven by RegCM4.0, with a 25 km × 25 km original spatial resolution, which was released by the National Climate Center of China [21]. The climate projection datasets have been revised on the national scales for China and are widely used to perform impact assessments of climate change across multiple different disciplines [39,40]. In addition, annual mean $CO_2$ concentration data for historical periods were obtained from the Scripps Institution of Oceanography (SIO). The projections of annual $CO_2$ concentration varied according to climate scenarios available from the IPCC Data Center. We used two representative concentration pathway (RCP) scenarios: RCP4.5 is a medium greenhouse gas emission scenario, whereas RCP8.5 is a high greenhouse gas emission scenario.

Figure 2 illustrates the recapitulative characteristics of climate change in the YREB. The annual mean air temperature was about 14.8 °C in the period 1986–2005. By the end of the 21st century, the mean temperature is projected to be 15.9 or 17.9 °C under the RCP4.5 or RCP8.5 scenario, respectively. The mean air temperature is projected to increase substantially from 2006 to 2099 (Figure 2a). Precipitation will slightly increase, by 3.4% compared with the present climate, towards the end of the 21st century (2070–2099) under RCP4.5. Similarly, precipitation will increase by 2.3% under the RCP8.5 scenario (Figure 2b). The atmosphere $CO_2$ concentrations continue to increase, and are projected to continuously increase during the period of 1960 to 2099. The $CO_2$ concentration will increase strongly under the RCP8.5 scenario; comparatively, it will increase moderately until 2070, and then fluctuate under the RCP4.5 scenario. The magnitudes of $CO_2$ concentration under RCP4.5 and RCP8.5 are estimated to be slightly over 500 and 900 ppm by the end of the 21st century, respectively (Figure 2c).

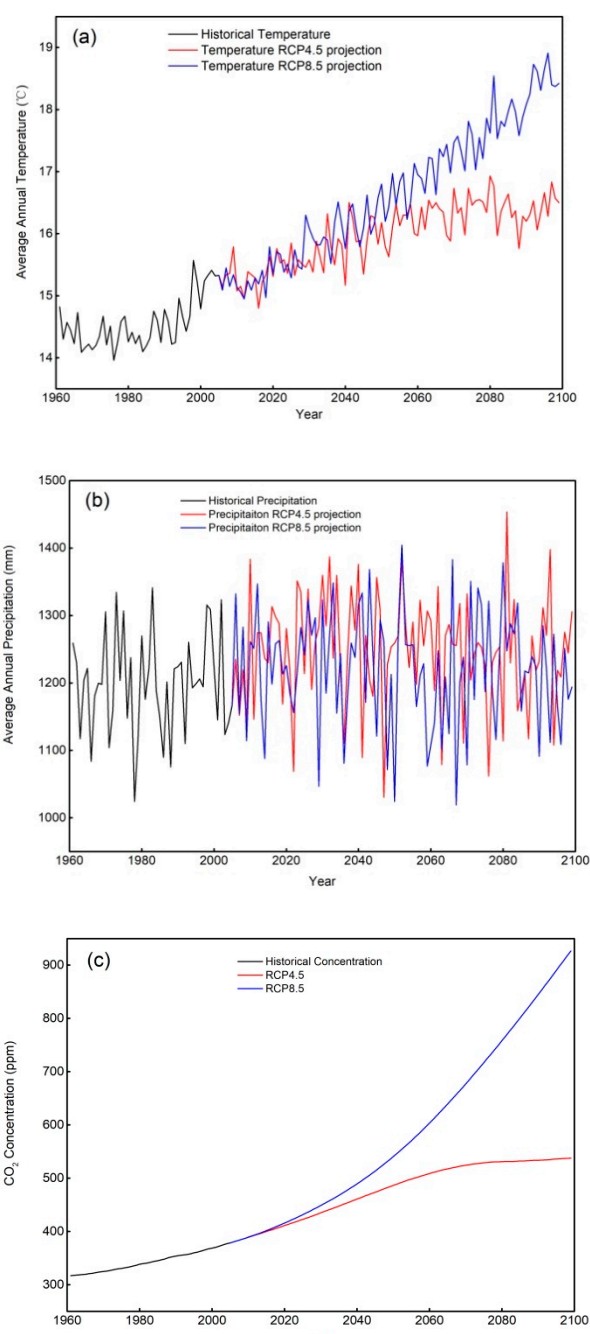

**Figure 2.** Annual variation in mean air temperature (**a**), precipitation, (**b**) and atmospheric $CO_2$ concentration (**c**) during the periods 1961–2005/2006–2099 under the RCP4.5 and RCP8.5 scenarios in the YREB.

*2.4. Simulations and Analysis*

Firstly, the equilibrium CEVSA2 model was performed by average climatic data and average $CO_2$ concentration in 1971–2000 until equilibrium status of the ecosystem was achieved, which aimed to eliminate the impacts of the initial ecosystem conditions on the simulation. Then, the dynamic model was run using transient climate data and the annual atmospheric $CO_2$ concentration during the period 1961–2099. The results of the dynamic simulation from 1986 to 2005 were used as a referenced for the current climate, and the outputs of 2020 to 2099 were analyzed to explore the spatiotemporal patterns and changing trends in vegetation NPP over the 21st century.

To show the spatiotemporal characteristics of vegetation NPP to climate change, we defined the period of 2020–2049 as the mid-term 21st century and the period of 2070–2099 as the late-term 21st century. In addition, because of the concerns of the impacts of not achieving global warming targets in key ecological regions, we also analyzed the impacts of 1.5 and 2 °C global warming targets on terrestrial NPP in the YREB. Here, based on the results from Jiang et al. (2016) [41], we extracted the projections of 1.5 and 2 °C warming under RCP4.5 and RCP8.5 scenarios, respectively. All representative periods in this study are shown in Table 1.

**Table 1.** The representative periods of the 21st century and the periods of 1.5 and 2 °C warming under RCP4.5 and RCP8.5 scenarios.

| Scenarios | Mid-Term | Late-Term | 1.5 °C Warming Period | 2 °C Warming Period |
|---|---|---|---|---|
| RCP4.5 | 2020–2049 | 2070–2099 | 2020–2039 | 2040–2059 |
| RCP8.5 | 2020–2049 | 2070–2099 | 2017–2036 | 2030–2049 |

To indicate the responses of vegetation NPP to climate change in the YREB, the percentage change of NPP (*R*) was defined as the normal value in different periods against its value in the respective reference period. Here, $NPP_{TP}$ is the mean NPP during the different target periods, and $NPP_{RP}$ is the mean NPP of the reference period. We adopted linear fitting to show the trend in annual NPP spatial and temporal patterns according to climate change under the two RCP scenarios. This was estimated using ordinary least squares based on unitary linear regression. Differences in the NPP slope at different warming targets were used to estimate the distinct responses of a further 0.5 °C increment of warming and the potential risks of plant growth to global warming in the YREB.

## 3. Results

### 3.1. Estimated Vegetation NPP in the Current Climate

The spatial distribution of plant NPP varied markedly over the entire YREB region, as shown in Figure 3. The annual NPP was estimated to be about $652.7 \pm 177.2$ g C m$^{-2}$ year$^{-1}$ during the reference period, which is similar to the results obtained by Miao et al., who used AVIM (674.1 g C m$^{-2}$ year$^{-1}$), and Luo et al., who used statistical data (687.4 g C m$^{-2}$ year$^{-1}$) [42], and the estimate was higher than that in the simulation performed by Zhang et al., who used the LPJ model (530.4 g C m$^{-2}$ year$^{-1}$), and the evaluation carried out by Wu et al., who used remote sensing data (472.6 g C m$^{-2}$ year$^{-1}$). The annual NPP in the YREB is about 1.7 times greater than that of the whole country, the total NPP of the YREB comprises about 33% that of the whole country, although the land area only accounts for about 21% of the total area. Spatially, the higher NPP values, above 750 g C m$^{-2}$ year$^{-1}$ in the YREB, are mainly located in the middle and lower reaches, which account for about 35.5% of the total area of the YREB, covering Jiangxi, Hunan, and Zhejiang provinces, and in some regions of the upper reaches, such as Chongqing municipality, Sichuan Province, and southern Yunnan Province. The lower NPP values, of less than 450 g C m$^{-2}$ year$^{-1}$, are distributed in the northwest of the YREB, which accounts for approximately 13.8% of the total area; these areas are mostly mountainous and plateau areas, including the northwest Sichuan Province and the northern Yunnan Province. Temporally, the tendency of annual mean NPP in the reference period showed significant growth ($R^2 = 0.71$, $p < 0.01$), which was consistent with the changing tendency of annual NPP on the national scale; however, the growth rate in the YREB was approximately 1.3 times higher than that on the national scale.

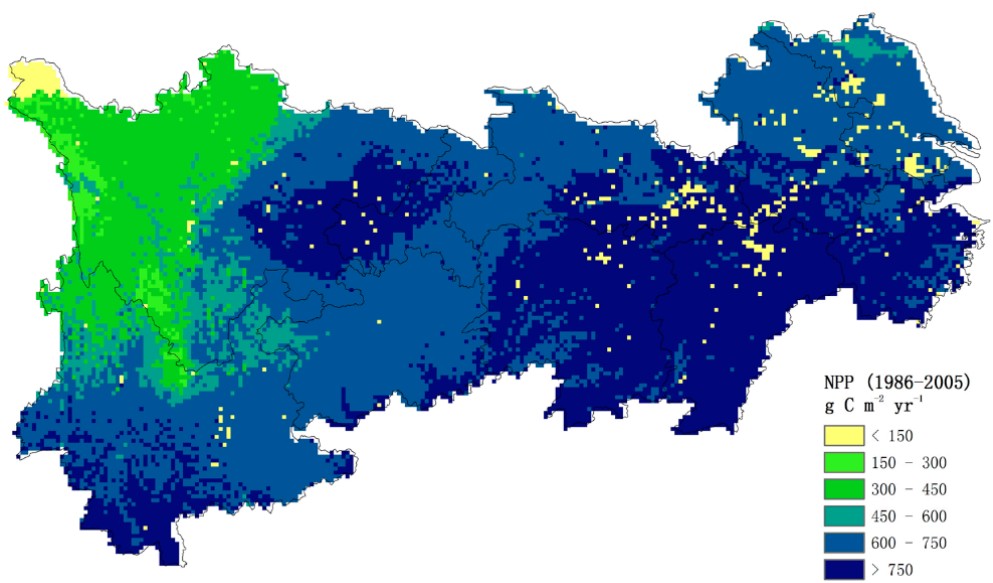

**Figure 3.** Spatial pattern of NPP in the reference period of 1986–2005 in the YREB.

### 3.2. Projected Vegetation NPP Due to Future Climate Change

Vegetation NPP is projected to continuously increase in the YREB in the 21st century. The annual mean NPP will increase to $738.7 \pm 190.7$ g C m$^{-2}$ year$^{-1}$ and $805.6 \pm 203.2$ g C m$^{-2}$ year$^{-1}$ under the RCP4.5 scenario, and $756.6 \pm 193.9$ g C m$^{-2}$ year$^{-1}$ and $924.8 \pm 241.6$ g C m$^{-2}$ year$^{-1}$ under the RCP8.5 scenario in the mid-term and late-term 21st century, respectively. Annual mean NPP was significantly correlated with the mean air temperature (RCP4.5: $R^2 = 0.72$, $p < 0.01$; ECP8.5: $R^2 = 0.93$, $p < 0.01$) for both RCP scenarios (Figure 4). Meanwhile, the NPP growth rate will clearly be higher under the RCP8.5 scenario than that under the RCP4.5 scenario, although both NPP growth rates are higher than that in the reference period.

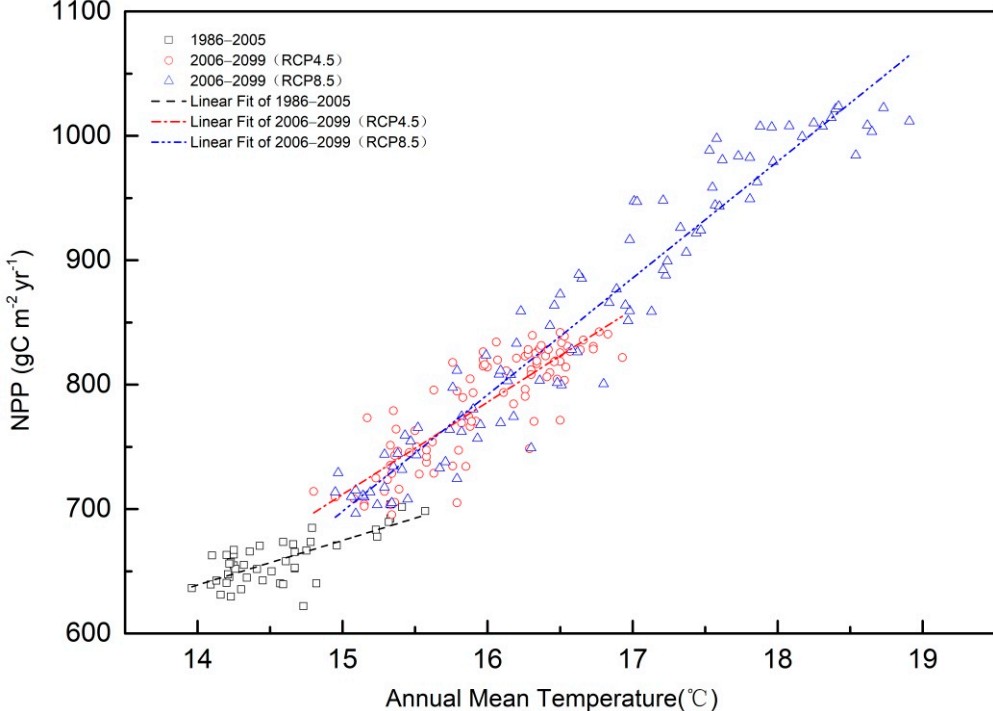

**Figure 4.** Effects on vegetation NPP and the growth rates of mean air temperature under different scenarios in the YREB.

The trends in annual NPP under the different scenarios indicate that there is a turning point of NPP changing to temperature increase. Before the average temperature rises to 15.7 °C, the rate of NPP change under RCP4.5 is shown to be higher than that under RCP8.5. As the mean air temperature continues to increase, NPP also keeps increasing under the high emission scenario. However, when the mean temperature rises above 17.5 °C under the RCP8.5 scenario, the value of NPP fluctuates at around 1000 g C m$^{-2}$ year$^{-1}$, and the trend in NPP change also shows a remarkable slowing down.

Although global warming prompts vegetation NPP to increase remarkably, our results also reveal that warming does not always have positive effects on the NPP. The interdecadal NPP trend shows that the growth rate of vegetation NPP appears to decrease after 2060 under RCP4.5, and slows down after 2070 under RCP8.5 (Figure 5). In addition, the atmospheric $CO_2$ concentration shows a positive effect on vegetation NPP based on the interdecadal trends of NPP; the NPP growth rate is consistently greater under higher atmospheric $CO_2$ concentrations, whether at the same warming level or in the same period. Similarly, the magnitudes and growth rates of NPP under higher $CO_2$ concentrations are greater than those under low $CO_2$ concentrations according to the interdecadal tendency of annual NPP.

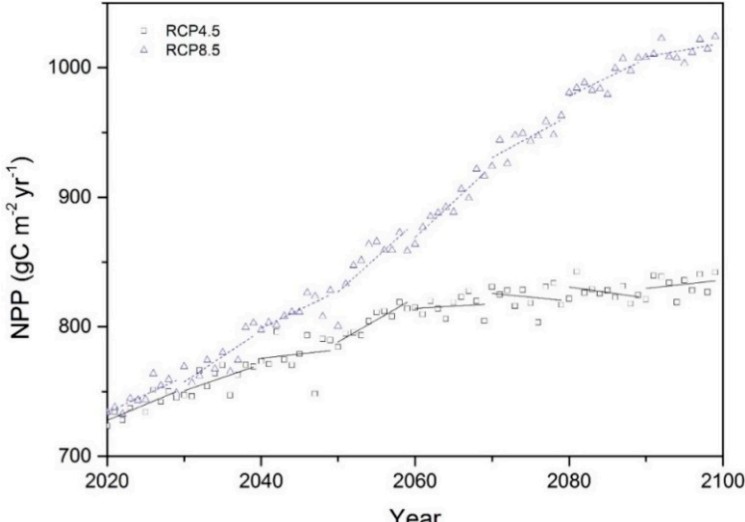

**Figure 5.** The annual mean NPP under the RCP4.5 and RCP8.5 scenarios during 2020−2099 in the YREB.

*3.3. Spatial Responses of NPP to Future Climate Change*

Figure 5 shows the spatial patterns of NPP variations in the mid-term and late-term 21st century under the two studied RCPs against to the reference period. Spatially, the increments of vegetation NPP in the mid-term period demonstrate a basic pattern of high in the west and low in the east under different RCP scenarios. However, in the late term, the spatial patterns of vegetation NPP show great difference under the RCP8.5 scenario. Generally, the overwhelming majority of areas show an increase in plant NPP in the future; fewer than 1% of the grids show a decrease in NPP, with the exception being that in the late term under the RCP8.5 scenario.

In the mid-term 21st century, vegetation NPP is projected to increase all over the region, the spatial patterns of NPP change will be similar for both RCP scenarios, and the greatest increase in vegetation NPP will be located in the Yunnan and Sichuan provinces, which both have lower NPP values under contemporary climate conditions (Figure 6). The projected vegetation NPP will increases by about 13.5% and 16.4% under the RCP4.5 and RCP8.5 scenarios, respectively. In the late-term 21st century, the magnitude of NPP change in most areas of the YREB will be 20% greater than that in the reference period under the same RCPs. Specifically, NPP will increase by 23% and 35% under the RCP4.5 and RCP8.5

scenario, respectively. Similarly, the magnitude of NPP change is projected to be higher under the RCP8.5 scenario than that under the RCP4.5 scenario. However, vegetation NPP is projected to decline in some areas under the RCP8.5 scenario in the late-term 21st century, which will be concentrated in the central–eastern region of the YREB, including Jiangxi, Hunan, and Hubei, comprising 3.6% of the entire YREB area.

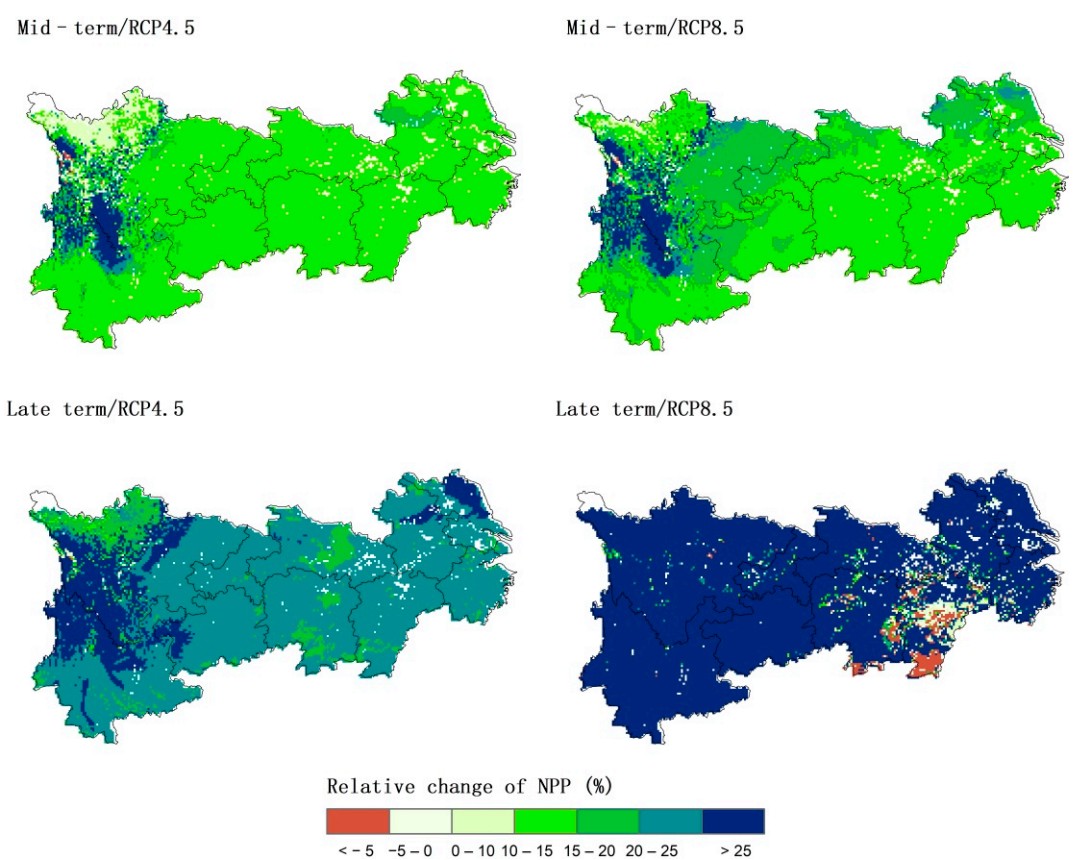

**Figure 6.** Spatial change patterns of NPP in the mid−term and the late−term 21st century under the RCP4.5 and RCP8.5 scenarios compared with the reference period in the YREB.

### 3.4. Vegetation NPP Responses to Global Warming

As shown in Figure 7, the spatial change in vegetation NPP presents similar patterns for both 1.5 and 2 °C global warming targets for different RCP scenarios. The magnitude of vegetation NPP increases considerably more in the western YREB than that in the east. Vegetation NPP is projected to increase strongly at both warming targets over the region; fewer than 1% of grids showed it to decrease compared with the reference period. The annual mean NPP is projected to increase to $730 \pm 188.1$ g C m$^{-2}$ year$^{-1}$ at 1.5 °C warming and to $771 \pm 198.3$ g C m$^{-2}$ year$^{-1}$ at 2 °C warming. The increase in NPP is approximately 14.7% at 1.5 °C warming and 21% at 2 °C warming compared with the reference period. Additionally, there is no significant difference in NPP variation under different scenarios, although vegetation NPP will increase by approximately 6% more in the 2 °C global warming period than in the 1.5 °C warming period. Generally, warming is projected to promote NPP even at 2 °C warming targets, and the additional 0.5 °C warming shows positive contributions to NPP increasing, especially under the RCP8.5 scenario.

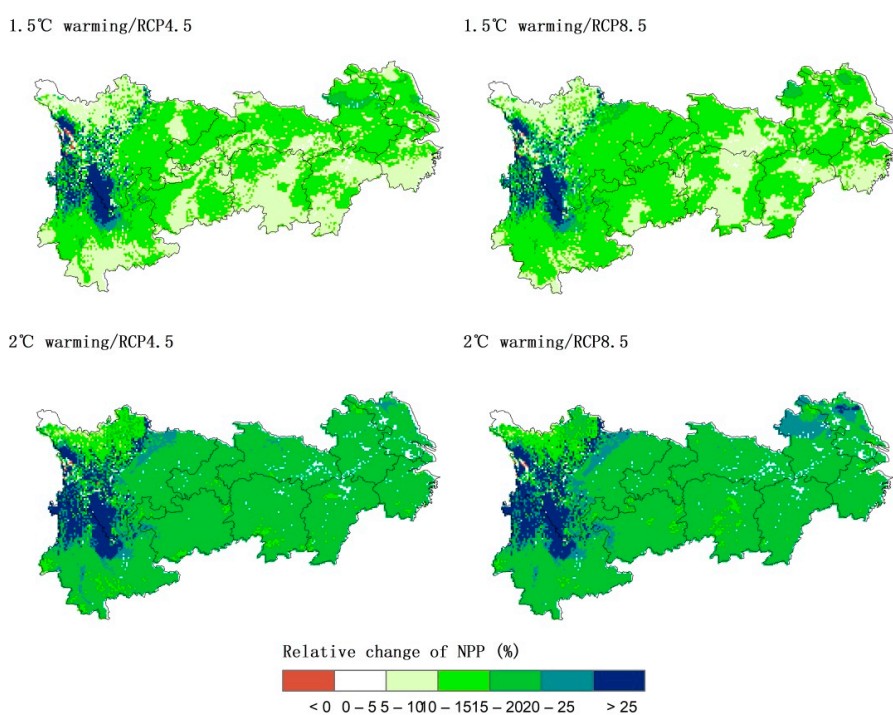

**Figure 7.** Spatial patterns of vegetation NPP at 1.5 and 2 °C global warming under the RCP4.5 and RCP8.5 scenarios in the YREB.

Although plant NPP is projected to increase in most areas of the YREB with the two global warming targets, the extra 0.5 °C warming will show a substantial impact on the growth rate of vegetation NPP. Figure 8 shows the different NPP tendencies in the 1.5 and 2 °C warming periods under different scenarios. A further 0.5 °C of warming would reduce the growth rate of NPP in about 31% of the regional areas under RCP4.5. Spatially, the growth rate of NPP in about one-third of the YREB, including places such as Sichuan, Yunnan, Guizhou, northern Hubei and Jiangsu, and northwestern Anhui, will be reduced at an enhanced warming target. Conversely, the trend in vegetation NPP is projected to keep growing in over 89% of the region at 2 °C warming under the RCP8.5 scenario. In most places, such as northern Yunnan, northeastern Sichuan, and northern Jiangsu and Anhui, there will be an even further increased growth rate of NPP at 2 °C warming.

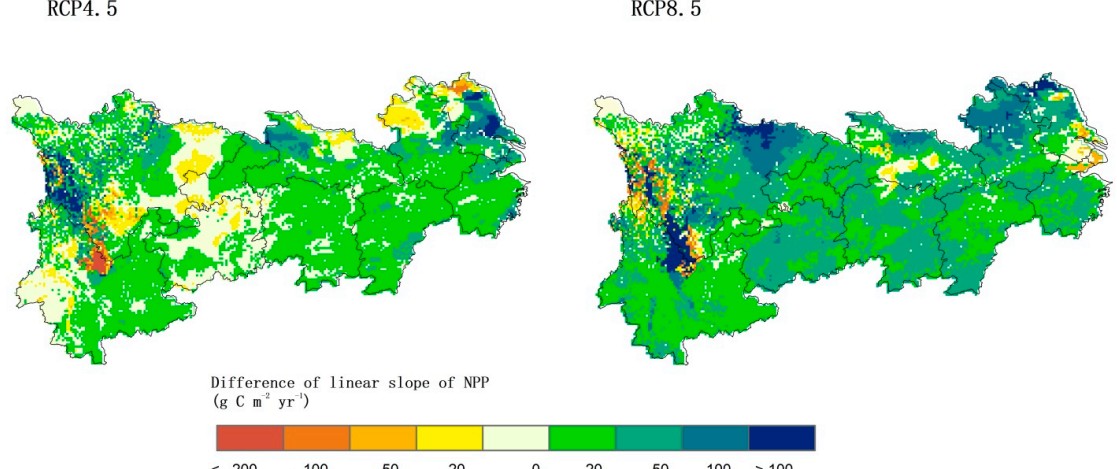

**Figure 8.** Spatial patterns of NPP trends at 1.5 and 2 °C global warming targets under the RCP4.5 and RCP8.5 scenarios.

## 4. Discussion

### 4.1. The Effects of Warming

Global warming is one of the most pervasive characteristics of climate change. Continuous warming has been proven to increase terrestrial vegetation NPP in recent decades; it was projected to increase in the 21st century in mid- and high- latitude regions, as well as alpine regions [1,43]. Previous studies in China also indicated that the rising meant air temperature was the dominant factor accounting for NPP changes on the national scale [14,36]. Based on the projection of high-resolution regional climate models, warming will also be widespread throughout the 21st century in the YREB (Figure 9). Our simulation results show that vegetation NPP is projected to increases in most parts of the YREB. Furthermore, the magnitude of NPP increments will be greater in the west than in the eastern YREB, which is associated with the differences in mean air temperature due to the higher altitude in this region. Our results are similar to those from research on alpine regions in southwestern China, such as the Yunnan–Guizhou Plateau and the Western Sichuan Basin [28]. The effects of different global warming targets on vegetation NPP also show that a rising temperature will promote increased vegetation productivity, which will be beneficial for terrestrial ecosystem carbon storage and achieving carbon neutrality on the regional scale.

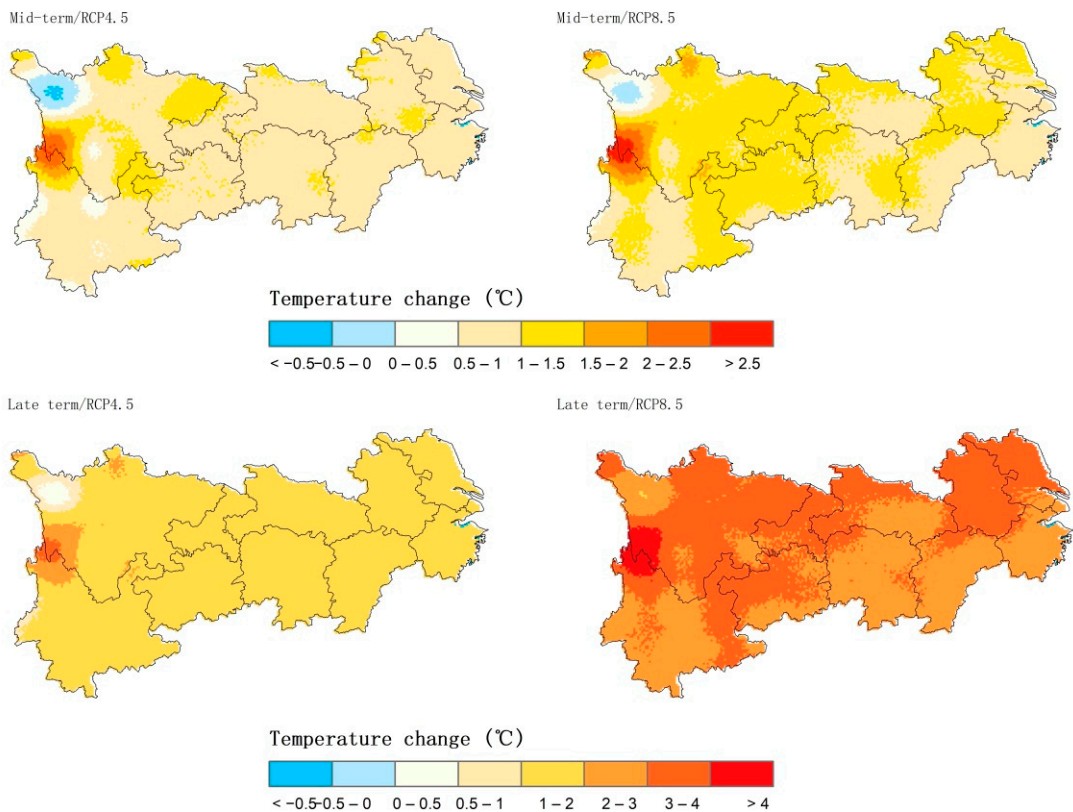

**Figure 9.** Mean air temperature change patterns in the mid–term and late–term 21st century compared with the reference period under the RCP4.5 and RCP8.5 scenarios.

### 4.2. The Effects of Precipitation Changes

As one of the most abundant water resource regions in China, the water supply for plant growth in the YREB is usually sufficient, and water resources rarely constrain vegetation productivity [44]. Based on the projections of regional climate models, there are no obvious trends in annual precipitation over the region (Figure 2b), although the variation in precipitation exhibits reasonable spatial heterogeneity, with high uncertainty (Figure 10). Precipitation is projected to increase in the west but generally decrease in the southern and

central YREB, and coupled with temperature changes, the spatial hydrothermal patterns will be greatly altered in this region.

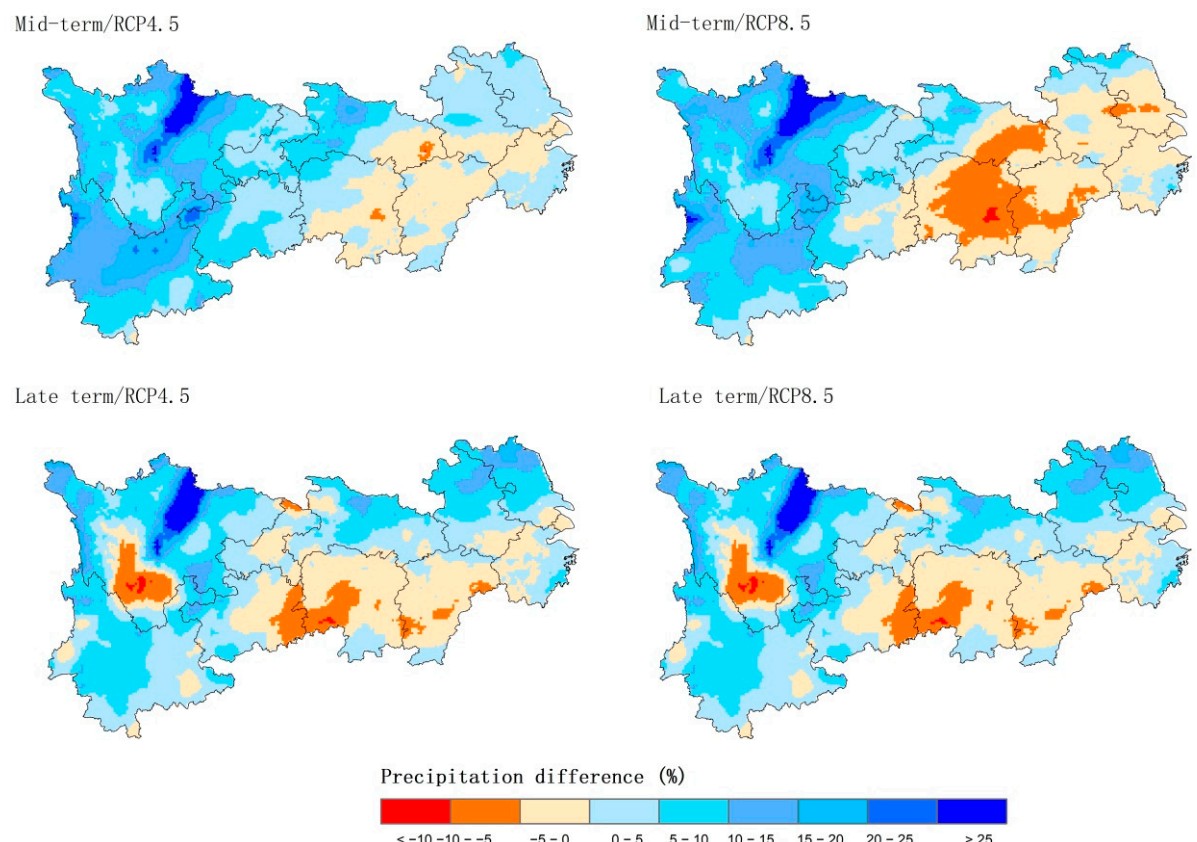

**Figure 10.** Spatial pattern of annual mean precipitation changes in the mid-term and late-term 21st century relative to the reference period under the RCP4.5 and RCP8.5 scenarios in the YREB.

The detrend fluctuation analysis showed that temperature and precipitation both exhibit positive correlations with the interannual variability of NPP (IAV NPP) during the period of 2021–2099 under RCP4.5, whereas under RCP8.5, only precipitation shows a positive correlation with IVA NPP (Table 2). This suggests that precipitation has a more dominant influence on IVA NPP than temperature, especially under higher emission scenarios, which indicates that as plant demand for water supply increases with rising temperatures, water resources may become a restricted factor for plant growth in this region. Thus, enhanced warming due to anthropogenic $CO_2$ emission promotes the growth and productivity of plants, but a sufficient water supply is necessary; otherwise, it may lead to stunted vegetation growth and reduced productivity. Vegetation NPP is projected to decline in the central–southern region of the YREB under RCP8.5 in the late-term 21st century maybe for this reason. Studies on other areas of China also found that declines in vegetation NPP will occur in water-deficit areas associated with rising air temperatures [7,45]. Therefore, water availability to terrestrial vegetation is particularly important in the YREB, especially so towards the end of 21st century. However, plant growth and vegetation NPP are always affected by integrated climatic conditions and other environmental factors; thus, the trends and fluctuation in NPP will respond in accordance with certain regional features as well as species diversity and complexity.

**Table 2.** The coefficients of detrended temperature and precipitation anomalies with IAV NPP under the RCP4.5 and RCP8.5 scenarios in the different periods of 21st century. Statistically significant correlations are marked with ** ($p < 0.01$) and * ($p < 0.05$).

|  |  | 2021–2049 | 2070–2099 | 2021–2099 |
|---|---|---|---|---|
| RCP4.5 | T | 0.04 | 0.61 ** | 0.39 ** |
|  | P | 0.55 ** | 0.35 | 0.38 ** |
| RCP8.5 | T | 0.27 | −0.01 | 0.15 |
|  | P | 0.58 ** | 0.52 * | 0.52 * |

*4.3. Factors Other Than Climate*

Except for temperature and precipitation, the elevated atmospheric $CO_2$ concentrations are a remarkable feature of climate change. The $CO_2$ concentration will increase even more under the high-emission scenario, and it will be more than twice the current level under RCP8.5 by the end of the 21st century (Figure 2c). Previous studies have confirmed that the fertilization effect of $CO_2$ contributes to the increase in vegetation NPP, but high $CO_2$ concentrations will indirectly have a negative impact on biodiversity, which may impact the stability of terrestrial vegetation productivity; thus, the effects of rising atmosphere $CO_2$ levels are still highly uncertainty [8,46,47]. Our results show that annual NPP and its changing rate under tRCP8.5 are greater than those under RCP4.5, especially at 4 °C warming (Figure 9). Higher $CO_2$ concentrations and warmer temperature bring further increases in NPP, which means that the elevated $CO_2$ concentrations will stimulate plant growth as well as temperature increases [48]. Vice versa, the effect of rising air temperature on vegetation would be limited without the cooperative influence of higher $CO_2$ concentrations. Our results suggest that the increase in $CO_2$ concentration has a significantly positive effect on NPP. Similar results by Zhu et al. (2018) [43] also indicated that the elevated atmospheric $CO_2$ concentration is the dominant factor on increasing NPP, particularly in mid- and low-latitude regions.

Land use and land cover change (LUCC) present realistic pressures on terrestrial ecosystems in addition to the impacts of climate change, and also have notable impacts on species and productivity [49,50]. Anthropogenic LUCC is a substantial factor that impacts terrestrial ecosystem functions and services associated with biodiversity and productivity, especially in areas with extensive human activity [17]. In this study, the CEVSA2 model did not consider the effects of LUCC, which would bring uncertainty in the projection. In the YREB, urban and construction areas have continuously increased in recent decades [51,52]. Additionally, wetlands have been shrinking since the 1970s [53]. However, forestland and grassland have shown an increasing tendency since the 1990s, due to a series of major ecological conservation and restoration projects, such as "the Grain for Green Program" and "Natural Forest Protection Projects". Now, there are high forest coverage rates in the YREB, significantly above the national average [25,51,54]. Some studies have suggested that the increases in NPP due to climate change are not enough to offset the decreases in plant NPP due to anthropogenic LUCC [55], although other studies have suggested that LUCC has already brought benefits for vegetation and terrestrial ecosystems in the YREB, for ecological restoration and protection projects in this region [26,56]. Furthermore, these ecological projects have been implemented and the effects have continually been improved, with the target of carbon neutrality. Therefore, ecosystem protection combined with climate change would be more conducive to the promotion of vegetation NPP. Thus, it would be expected that the terrestrial carbon storage in this region has great potential to improve, especially for the forest ecosystem [57]. Another study found that the carbon sequestration by terrestrial ecosystems in the YREB is projected to comprise about 42% of China's total capacity by the end of the 21st century; the magnitude will be higher because of ecological policies and projects in this region [37]. However, LUCC intensively affects ecosystems, including species, vegetation types, biodiversity, and productivity, and it is a great challenge to simulate the effects of both anthropogenic LUCC and climate change in ecosystem models [1]. Model development is ongoing and could consider LUCC effects by

coupling a module of climate change with land use, and it could consider the impacts of human management measures on terrestrial ecosystems, such as natural forest protection and the management of nature reserves, in order to reduce uncertainties in modeling vegetation growth on the regional scale.

## 5. Conclusions

In this study, we used a process-based ecosystem model, CEVSA2, driven by high-resolution climate data, to project future vegetation NPP, analyze the spatial and temporal changing patterns of NPP, and explore the responses of NPP to key climate factors in the YREB. Our study indicates that global warming will promote vegetation NPP, and vegetation NPP will increase strongly in the 21st century in the YREB; it is estimated to increase by about 14% under RCP4.5 and 16% under RCP8.5 in the mid-term, and 23% under RCP4.5 and 35% under RCP8.5 in the late-term. Generally, NPP is projected to be higher under the RCP8.5 scenario than that under the RCP4.5 scenario, associated with the positive effects of high atmospheric $CO_2$ concentrations. NPP is projected to increase at both global warming targets as well. The magnitude of the NPP increase will be approximately 15% at 1.5 °C and 21% at 2 °C global warming compared with the reference period, which are higher values than those expected on the national scale. However, continuous warming does not always result in positive contributions to vegetation NPP; the trends and variations in NPP change show considerable differences for different warming levels and climate change scenarios. In addition, precipitation exerts a more dominant role in NPP changes than air temperature under the higher emission scenario, especially towards the end of the 21st century. Our findings on vegetation NPP change and its responses to climate change suggest that climate change may be beneficial for vegetation growth and the carbon storage of terrestrial ecosystems in the YREB, but the projections of and responses to NPP exhibit high uncertainty and complexity. Our research contributes a detailed modeling study on a significant river-basin scale, which covers diverse terrestrial ecosystems and rich biodiversity. Our conclusions contribute to the knowledge of sustainable forest management and terrestrial ecosystem adaptations to climate change on river-basin scales and in subtropical monsoon zones.

**Author Contributions:** Conceptualization, L.Y. and F.G.; methodology, L.Y. and B.Z.; software, F.G. and L.Y.; validation, Y.X. and Y.Z.; formal analysis, L.Y.; writing, L.Y. and Y.X.; review and editing, Y.Z. and Y.X.; visualization, L.Y.; supervision, L.Y. and B.Z. All authors have read and agreed to the published version of the manuscript.

**Funding:** This research was funded by National Natural Science Foundation of China grant number [41991285].

**Institutional Review Board Statement:** Not applicable.

**Data Availability Statement:** Data are available from the authors upon reasonable request as the data need further use.

**Acknowledgments:** We acknowledge Shi Ying provided the projection climate data of RCM in China.

**Conflicts of Interest:** No conflict of interest exists in the submission of this manuscript, and the manuscript is approved by all authors for publication.

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
