# Peer review of "Projections of the Net Primary Production of Terrestrial Ecosystem and Spatiotemporal Responses to Climate Change in the Yangtze River Economic Belt"

_diversity, doi:10.3390/d14050327_

Round 1
Reviewer 1 Report
Dear authors,
the study is well performed and the methods and models that you use are adequate. But I have one major concern: You write for the journal "diversity"but you never consider that question in your contribution. That is a weakness. As we know, warming and increase of CO2-concentrations influence the competition between species to a high degree. Thus, we must expect that fastly-growing plants will get higher dominance and, probably, other species will decrease or disappear. Please, mention that question at least in discussion and conclusions.
Best regards
Your reviewer
Author Response
Dear reviewers and editors,
We appreciate the constructive advice of reviewers and efficient work of editors. We took care to incorporate all recommendations and conducted a careful modification and improvement. We think that the comments helped to improve the quality of the manuscript. Please find below our replies (red italics) to the comments and content added in manuscript.
Sincerely,
LI YU

Reviewer 2 Report
This is a useful study and suitable for publication, however the entire manuscript needs to be edited for better expression in English I believe the authors and editors should be resolve the problems through a robust editing process, which clarifies the expression used. Other than this, and after it has been comprehensively edited, I have no hesitation in recommending this MS as suitable for publication.
1. What is the main question addressed by the research? The research deals with the following questions:how net primary production (NPP) in the Yangtze Valley will change/is changing under climate change? and can this be usefully and credibly modelled?2. Do you consider the topic original or relevant in the field, and if so, why? The topic has global relevance because of the significance of NPP as driver of ecosystem processes and benefits, including for carbon sequestration, but also for bio-diversity conservation, and agriculture
3. What does it add to the subject area compared with other published material? The MS adds a detailed and current modelling study focused on a major river basin, which contains many areas of international significance. The river basin itself is global significant. The area spanned by the study is large and covers a significant proportion of China. The MS contributes to the existing published material because it tackles this question at the scale of a significant river basin which covers diverse landscapes/ecosystems
4. What specific improvements could the authors consider regarding the
methodology? The methodology appears to be sound, but there will always be scientific contestation about modelling methods and results, which the authors acknowledge
5. Are the conclusions consistent with the evidence and arguments
presented and do they address the main question posed? Yes the conclusions are consistent with the evidence provided and arguments mounted and they address the main question posed
6. Are the references appropriate? They are sufficient, comprehensive and cover off on the main topics well, including methods and methodological questions.
7. Please include any additional comments on the tables and figures. The figures are of a high standard and clear
Author Response

(The authors gave the same response as above.)

Round 2
Reviewer 1 Report
manuscript is now ready for publication
Reviewer 2 Report
MS is suitable for publication after the comprehensive editing